# Effect of SARS-CoV-2 S protein on the proteolytic cleavage of the epithelial Na+ channel ENaC

Germán Ricardo Magaña-Ávila[1,2], Erika Moreno[1], Consuelo Plata[1], Héctor Carbajal-Contreras[1,3], Adrian Rafael Murillo-de-Ozores[1,2], Kevin García-Ávila[1], Norma Vázquez[4], Maria Syed[5], Jan Wysocki[5], Daniel Batlle[5], Gerardo Gamba[1,3,4], María Castañeda-Bueno[1]*

1 Department of Nephrology and Mineral Metabolism, Instituto Nacional de Ciencias Médicas y Nutrición Salvador Zubirán, Mexico City, Mexico, 2 Facultad de Medicina, Universidad Nacional Autónoma de México, Mexico City, Mexico, 3 Facultad de Medicina, PECEM (MD/PhD), Universidad Nacional Autónoma de México, Mexico City, Mexico, 4 Instituto de Investigaciones Biomédicas, Molecular Physiology Unit, Universidad Nacional Autónoma de México, Mexico City, Mexico, 5 Department of Medicine, Division of Nephrology and Hypertension, Northwestern University Feinberg School of Medicine, Chicago, IL, United States of America

* maria.castanedab@incmnsz.mx, mcasta85@yahoo.com.mx

**Data Availability Statement:** All relevant data are within the paper and its Supporting Information files. Western blot replicates can be found with the

## Abstract

Severe cases of COVID-19 are characterized by development of acute respiratory distress syndrome (ARDS). Water accumulation in the lungs is thought to occur as consequence of an exaggerated inflammatory response. A possible mechanism could involve decreased activity of the epithelial Na+ channel, ENaC, expressed in type II pneumocytes. Reduced transepithelial Na+ reabsorption could contribute to lung edema due to reduced alveolar fluid clearance. This hypothesis is based on the observation of the presence of a novel furin cleavage site in the S protein of SARS-CoV-2 that is identical to the furin cleavage site present in the alpha subunit of ENaC. Proteolytic processing of αENaC by furin-like proteases is essential for channel activity. Thus, competition between S protein and αENaC for furin-mediated cleavage in SARS-CoV-2-infected cells may negatively affect channel activity. Here we present experimental evidence showing that coexpression of the S protein with ENaC in a cellular model reduces channel activity. In addition, we show that bidirectional competition for cleavage by furin-like proteases occurs between ⟨ENaC and S protein. In transgenic mice sensitive to lethal SARS-CoV-2, however, a significant decrease in gamma ENaC expression was not observed by immunostaining of lungs infected as shown by SARS-CoV2 nucleoprotein staining.

## Introduction

Severe acute respiratory syndrome coronavirus 2 (SARS-CoV-2) is a positive single stranded RNA virus known to be the causative agent of COVID-19 (Coronavirus disease 19). SARS-CoV-2 emerged in Wuhan, China in late 2019, and since then has spread throughout all

following link: https://figshare.com/articles/figure/SUPPORTING_INFORMATION_Maga_a-Avila_et_al_PLOS_ONE_2024/25199699.

**Funding:** The work was supported by grants No. 101720 and A1-S-8290 from Consejo Nacional de Humanidades, Ciencias y Tecnologías (CONAHCyT), Mexico to M-CB and GG, respectively. German R. Magaña Avila is a doctoral student from the "Programa de Doctorado en Ciencias Biomédicas, Universidad Nacional Autónoma de México (UNAM)" and received a fellowship from CONAHCyT (CVU 942671). We acknowledge the support of a gift from the Joseph and Bessie Feinberg Foundation, a National Institutes of Health grant (1R21 AI166940-01) and a the Northwestern University Clinical and Translational Sciences Institute (NUCATS) COVID-19 Collaborative Innovation Award to DB. The funders had no role in study design, data collection and analysis, decision to publish, or preparation of the manuscript.

**Competing interests:** During these studies, G. Gamba received unrelated support from the National Institutes of Health (grant DK51496). D. Batlle and J. Wysocki are coinventors of patents entitled "Active low molecular weight variants of angiotensin converting enzyme 2 (ACE2), "Soluble ACE2 variants and uses therefor." D. Batlle is founder of Angiotensin Theraputics, Inc. D. Batlle has received consulting fees from Advience and Traveri, all unrelated to this work. During these studies, D. Batlle received unrelated support from the National Institute of Diabetes and Digestive and Kidney Diseases (grant R01DK104785) and from a grant from AstraZeneca. D. Batlle also reports research funding from the Feinberg Foundation; J. Wysocki reports being a scientific advisor for Angiotensin Theraputics, Inc. All other authors have nothing to disclose. This does not alter our adherence to PLOS ONE policies on sharing data and materials.

countries of the world, infected over 700 million people, and caused over 6 million deaths. SARS-CoV-2 principally affects the respiratory system. In the most severe cases, acute respiratory distress syndrome (ARDS) develops, with acute onset of bilateral infiltrates, severe hypoxemia, lung edema, and a systemic inflammatory response known as cytokine release syndrome that leads to multiorgan failure [1,2].

The current understanding of cellular infection and organ tropism is based on evidence of the interaction between the viral surface protein Spike (S) and host cell´s membrane proteins and proteases that mediate cell entry. The S protein is a membrane bound glycoprotein forming a homotrimer that binds to the membrane-bound angiotensin converting enzyme 2 (ACE2) in host cells, through its receptor binding domain located in the S1 subunit [3,4]. The S1 and S2 subunits are produced by proteolytic cleavage of the full-length S protein. Once bound to the membrane, the S protein is cleaved at two positions, the S1/S2 site, located in the boundary between S1 and S2, and the S2' site, located several residues downstream within the S2 portion. Cleavage at both sites is crucial for viral entry. Proteolytical processing is mediated by membrane bound serine proteases such as TMPRSS2, cathepsins, and furin-like enzymes that are found in specific cell types such as type II pneumocytes, proximal tubule cells, arterial, and venous endothelial cells, and brain cells [3–5]. The co-expression of ACE2 with the molecular machinery for proteolytical processing determines the efficiency of viral infection.

At the early beginning of the pandemic several groups identified a multibasic motif at the S1/S2 cleavage site of the SARS-CoV-2 S protein that is not present in other closely related coronaviruses [4,6]. This motif matches the minimal sequence required for cleavage by furin [7]. Thus, Hoffman et al. tested the effect of a furin inhibitor on protein cleavage and observed that it was indeed prevented [6]. In addition, it was shown that this furin cleavage site was required for efficient S protein processing in human cells, as well as S protein-driven cell-cell fusion. Moreover, entry of S protein-expressing pseudotypes to human lung cells was dependent on the presence of the S1/S2 furin cleavage site. Interestingly, Johnson et al. later observed that the mutant SARS-CoV-2 virus carrying a deletion in the furin cleavage site (ΔPRRA) of the S protein was less pathogenic than the wild type virus in hamsters and mice [8].

In an *in silico* analysis, Anand et. al. identified that, among all proteins encoded in the human genome, the only one that contains a motif 100% identical to the furin cleavage site of SARS-CoV-2 S is the alpha subunit of the epithelial sodium channel (αENaC) [9]. ENaC sodium channels, heterotrimers conformed by alpha, beta and gamma subunits, mediate $Na^+$ fluxes across plasma membranes of a variety of epithelia, including renal tubular, distal colon, and respiratory epithelia. In the lower airways of the respiratory system, ENaC is co-expressed with ACE2 and furin in type II pneumocytes [9]. These cells mediate amiloride sensitive reabsorption that is the rate limiting step in alveolar fluid clearance [10,11]. Thus, ENaC function in the lung is crucial to establish a normal liquid-air interphase that enables gas exchange. This is evidenced, for example, by the phenotype of α-ENaC knockout mice that die within a few hours of birth from acute lung edema [12]. In addition, the function of ENaC in endothelial cells is also relevant for the maintenance of the alveolar-capillary barrier. Thus, in these cells, its inhibition has been proposed to contribute to alveolar edema [13,14].

ENaC activity is regulated by proteolytic processing of the alpha and gamma subunits. αENaC is cleaved at two positions by furin and perhaps other furin-like proteases in the trans-Golgi network, allowing for export of the channel to the plasma membrane. γENaC is cleaved at one site by furin and, later on, at a second site in the plasma membrane by a membrane-bound or extracellular protease [15,16]. These proteolytic events are necessary for full channel activation. A model frequently used for the study of ENaC is the X. laevis heterologous expression system, where furin-dependent activation of ENaC occurs [17]. In fact, this system was used to demonstrate that cleavage of α and γENaC is crucial for channel activation [18–20].

Given the importance of furin-mediated proteolytic processing for ENaC channel function and given the evidence that furin-like proteases participate in S protein processing in SARS--CoV-2 infected cells, it has been hypothesized [9,21–24] that the hijacking of furin-like proteases by the S protein in infected cells may affect ENaC processing and activity and that this may contribute to the pathogenesis of the disease. Thus, in the present work we experimentally tested this hypothesis in the *X. laevis* system and were able to demonstrate competition between the S protein and αENaC for cleavage by a furin-like protease. Furthermore, decreased ENaC cleavage in the presence of S protein led to reduced levels of channel activity. We did not observe, however, changes in γENaC expression levels by immunofluorescence in lungs from mice infected with a lethal dose of SARS-COV2. Thus, although *in vitro* data clearly shows competition between ENaC and S protein for furin-mediated cleavage, further work is necessary to demonstrate if ENaC function in type II pneumocytes is affected *in vivo* by SARS-CoV2 infection.

## Materials and methods

### Xenopus laevis heterologous expression system

The use of X. laevis for oocytes extraction was approved by the Animal Care and Use Committee of the Instituto Nacional de Ciencias Médicas y Nutrición Salvador Zubirán. The cRNA for oocyte injection was synthesized in vitro from linearized cDNA using the T7 and SP6 RNA polymerase mMESSAGE mMACHINE kits (Invitrogen). Oocytes were surgically extracted from Tricaine (0.17%) anesthetized adult female Xenopus laevis frogs and incubated in $Ca^{2+}$ free ND96 medium (96 mM NaCl, 2 mM KCl, 1 mM $MgCl_2$ and 5 mM HEPES, pH7.4) with type B collagenase for 1.5 hours. After four washes with ND96 medium oocytes were incubated at 16˚C in ND96 medium. The next day, oocytes were microinjected with 50 nl of $H_2O$ alone or containing cRNAs encoding alpha, beta, and gamma ENaC (from rat) in an equimolar proportion (0.5 μg/ul of each cRNA) and with 0.05–0.6 μg/ul cRNA encoding the wild type ($S^{wt}$) or mutant ($S^{\Delta PRRA}$) S protein.

Injected oocytes were maintained at 16˚C until the day of the experiment.

### Two-electrode voltage clamp

To assess the effects of S protein expression on ENaC activity, amiloride sensitive $Na^+$ currents were measured by two-electrode voltage clamp in *Xenopus laevis* oocytes. Measurements were made forty-eight hours after injection. Oocyte membrane currents were recorded using an OC-720C voltage clamp (Warner Instruments, Hamden, CT) filtered at 15 Hz, digitized, and recorded with the PATCH MASTER software (HEKA, Germany). Data were analyzed as previously described [25,26]. Oocytes were clamped at a holding potential ($V_h$) of 100 mV and amiloride-sensitive current amplitudes were obtained by determining the difference in current before and after addition of 10 μM amiloride to the bath. For periods when I-V protocols were not being run, the oocytes were clamped of −100 mV, and the current was monitored and recorded. I-V protocols consisted of 150 ms at $V_h$ followed by 900-ms of 20-mV steps from $V_h$ to −140 mV and +60 mV, ending with 150 ms at $V_h$. The I-V protocols were run in a 110 mM $Na^+$ solution (mM: 110 NaCl, 2 $CaCl_2$, 10 mM HEPES, pH 7.4). All experiments were performed at room temperature. The oocytes were bathed 1–3 min with the test solution before the I-V protocol was ran. For statistical analysis, two-way ANOVA and Tukey post hoc tests were performed to determine if the I-V curves were significantly different.

## Immunoblots

Forty-eight hours after injection, oocytes were lysed, protein concentration was quantified, and lysates were separated by SDS-PAGE. Proteins were electrotransferred onto PVDF membranes and immunoblotting was performed. Bound antibodies were detected by chemiluminescence using the Luminata Forte Western HRP substrate (Millipore). For more details see supplementary materials.

## Animal experiments

All work with live SARS-CoV-2 was performed in the BSL-3 facility of the Ricketts Regional Biocontainment Laboratory, according to a protocol approved by the Institutional Animal Care and Use Committees of Northwestern University (approval number IS00004795) and University of Chicago (approval number 72642).

Ten K18hACE2 transgenic mice (5 male and 5 female) were inoculated with $2 \times 10^4$ PFU of SARS-CoV-2 (Washington strain) intranasally as previously reported by us [27]. Animals were weighed once daily and monitored twice daily for health using a clinical scoring system [27]. Animals that had lost >20% of their body weight or had a severely worsened clinical score (>3) were humanely euthanized which was considered as fatal event. Based on the severity of clinical score, all mice had to be euthanized on days 6–7. The lung was taken from all animals and fixed in formalin. Formalin-fixed lungs were released from the BSL-3 facility after verifying absence of infectious virus and were then paraffin embedded and cut to slides (4 μm) by the Mouse Histology and Phenotyping Laboratory Center, Northwestern University. We then performed lung tissue staining for ENaC from eight out of ten SARS-Cov-2-infected mice that had still enough tissue available for this study and compared it with the staining of lungs obtained from three uninfected control mice.

## Immunofluorescence staining of ENaC in k18hACE2 mice infected with SARS-CoV-2

Tissue sections were deparaffinized and rehydrated. Antigen retrieval of the sections was performed by using a retrieval buffer (pH 6.0) in a microwave according to standard protocol. Sections were then rehydrated and washed with PBS. Sections were permeabilized with 0.3% Triton-100. Blocking of non-specific binding was done by incubating tissue sections with 10% BSA in PBS. Primary antibodies were diluted in a solution containing 5% BSA in the following dilutions: anti-ENaC antibody (StressMarq, SPC-405, Rabbit, 1:400) and anti-Surfactant Protein A (SFTPA1) antibody (antibody.com, A83038, Goat, 1:150). Negative control was carried out by adding a solution in which the primary antibodies were diluted but without the primary antibody. All sections were incubated in a humidified chamber at 4C overnight. Sections were then washed with PBS. Then, secondary antibodies were added (Alexa donkey anti-rabbit 647 (1:100) and Alexa donkey anti-goat 488 (1:150), respectively) and incubated in darkness for 60 mins. After washing with PBS-Tween 20, ProLong Gold anti-fade solution was added and covered with a coverslip. Slides were examined under a Zeiss confocal microscope. See γENaC antibody validation data in Figs 2 and 3 in S1 File. It must be noted that SFTPA1 can also be secreted by non-ciliated bronchiolar cells, submucosal gland and epithelial cells of other respiratory tissues, such as the trachea and bronchi [28]. However, we did co-staining studies in the context of alveoli and not the airway tissues, such as bronchioles, bronchi, or trachea where SFTPA1 can also be present. Since those structures are easily discernable from the alveolar space under microscopy, we think that this should not affect the interpretation of our co-staining images that were focused on the alveolar space and not on airways.

See additional Materials and Methods in S1 File.

## Results

### Coexpression of the S protein with ENaC reduces proteolytic processing of the alpha subunit of the channel in a cellular system

To explore whether S protein expression affects αENaC processing by furin, *X. laevis* oocytes were injected with the cRNA encoding for the α, β, and © subunits of ENaC, in the absence or presence of wild type S protein ($S^{WT}$) or an S protein mutant harbouring a deletion in the furin cleavage site ($S^{\Delta PRRA}$). In oocytes injected with the ENaC subunits only, the full-length form of αENaC, as well as the C-terminal cleaved form of αENaC, were observed in immunoblots performed with an antibody against the HA C-terminal tag. Interestingly however, when ENaC was co-expressed with $S^{WT}$ at different concentrations, the cleaved form of αENaC was no longer detected in the blots even in samples from oocytes injected with the lowest amount of $S^{WT}$ cRNA (Fig 1). In contrast, higher amounts $S^{\Delta PRRA}$ expression were necessary to achieve equivalent reductions in the cleaved form of αENaC.

The proteolytic processing of $S^{WT}$ was also appreciated in the immunoblots. As expected, this processing was not observed for the $S^{\Delta PRRA}$ mutant. It is noteworthy that in the presence of ENaC, $S^{WT}$ processing was reduced (compare lane 2 with lane 7 of the blot in Fig 1 in which equivalent levels of $S^{WT}$ cRNA were injected). When a fixed amount of $S^{WT}$ protein was expressed in the presence of different combinations of ENaC subunits it was appreciated that only in the presence of the three ENaC subunits $S^{WT}$ protein cleavage was inhibited. Of note, as previously reported, only in the presence of the three ENaC subunits was αENaC and γENaC observed. Thus, only under conditions in which ENaC cleavage was observed, inhibition of $S^{WT}$ protein cleavage occurred. Moreover, in the presence of increasing amounts of γENaC, cleavage of αENaC increased, but cleavage of $S^{WT}$ protein decreased. These results suggest that competition between $S^{WT}$ and ENaC for furin-mediated cleavage occurs in this cellular system.

### ENaC activity decreases when $S^{WT}$ is coexpressed in X. laevis oocytes

We measured amiloride-sensitive $Na^+$ currents in oocytes expressing functional ENaC channels. As shown in Fig 2, ENaC-injected oocytes exhibited significantly higher amiloride-sensitive currents than water-injected oocytes, with the previously described positive reverse voltage for this channel [25,26]. Interestingly, as previously reported [29], co-expression $S^{WT}$ with ENaC, reduced ENaC currents (Figs 1 and 2 in S1 File). The observed decrease in αENaC proteolytic processing in $S^{WT}$-expressing oocytes (Fig 1) may be responsible for the decreased channel activity under these conditions. In contrast, no significant difference was observed between oocytes expressing only ENaC and oocytes expressing ENaC plus the $S^{\Delta PRRA}$ mutant. However, others have reported that a similar S mutant exerts a smaller inhibitory effect on ENaC currents than the wild type protein, but still exerts a significant inhibitory effect, which could be explained by the prevention of ENaC cleavage that we report [29].

### Alterations in expression levels of ENaC are not evident through immunostaining of lungs from SARS-CoV2- infected mice

For analysis of ENaC expression in lungs from control mice and mice infected with SARS-CoV2, immunofluorescent staining was performed (Fig 3). Validation tests for γENaC antibody were performed in lung and kidney tissue (Figs 2 and 3 in S1 File) and this antibody has

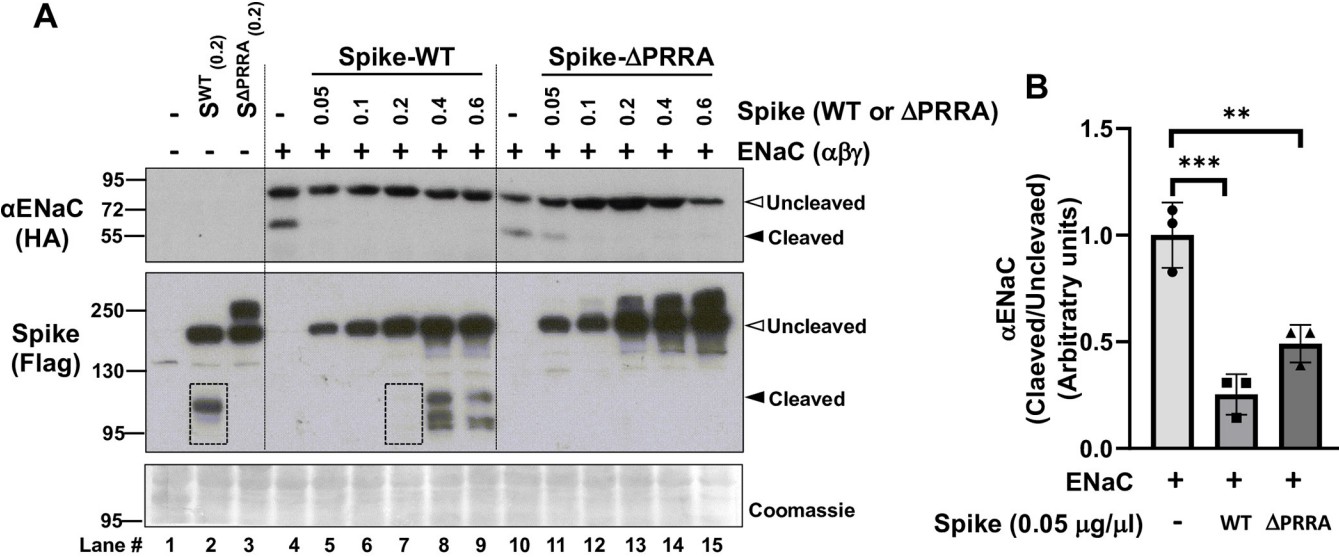

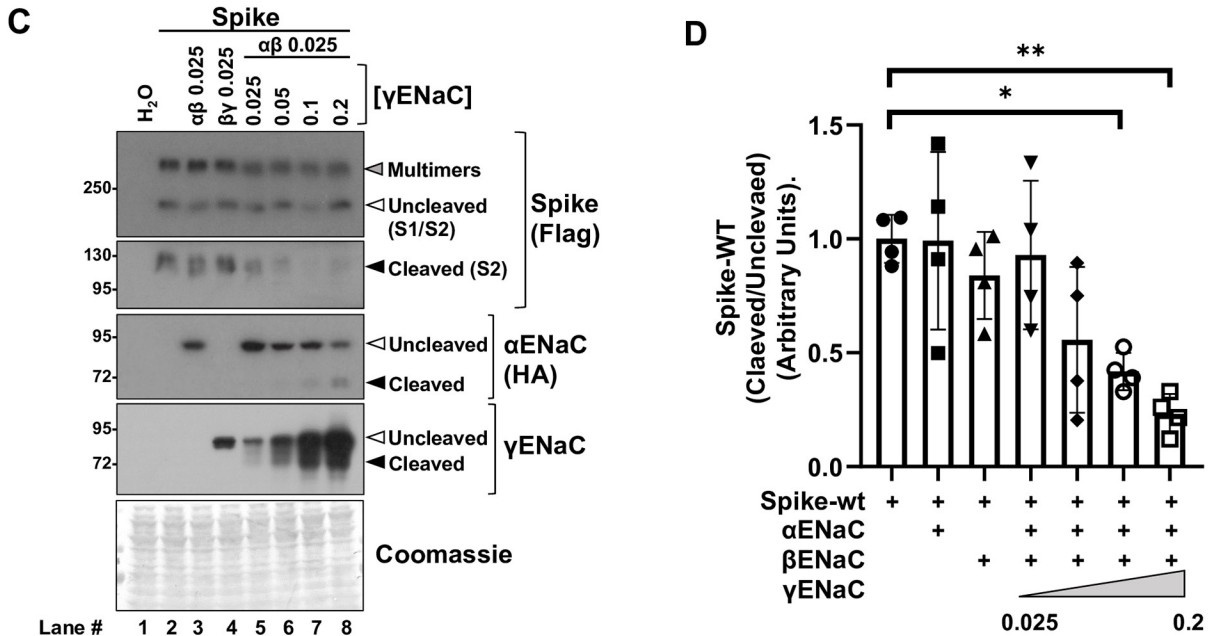

**Fig 1. αENaC and S protein compete for furin-mediated cleavage in a cellular system.** (A) αENaC processing in the X. laevis oocyte expression system was assessed by immunoblot. The effect on αENaC processing of S protein coexpression was explored by coinjecting increasing amounts of S protein cRNA with the ENaC subunits cRNA. The band corresponding to cleaved αENaC (65 kDa) was clearly observed in the absence of S protein expression, however, this band was only barely detected in samples from oocytes injected with the lowest amounts of S protein cRNA. Coexpression of the S protein furin cleavage site mutant ($S^{\Delta PRRA}$) with ENaC also negatively affected αENaC processing, although slightly higher amounts of expressed $S^{\Delta PRRA}$ protein were necessary to observe similarly reduced levels of ENaC processing than with $S^{WT}$. In the bottom blot the proteolytic processing of $S^{WT}$, but not $S^{\Delta PRRA}$, is observed, as expected. It is noteworthy that proteolytic processing of $S^{WT}$ was clearly prevented in the presence of ENaC coexpression (bands corresponding to the cleaved S2 subunit observed in oocytes injected with the same amount of $S^{WT}$ in the absence or presence of ENaC are highlighted with boxes). (B) Results of quantitation of the band corresponding to the cleaved form of αENaC of the immunoblots represented in A. Data are mean ± SEM, **$p < 0.01$, ***$p < 0.001$ vs. the only ENaC group, n = 3. ANOVA followed by Tukey tests were performed. (C) Effect of expression of different combinations of ENaC subunits on $S^{WT}$ protein cleavage. $S^{WT}$ protein cRNA was injected at a constant amount and co-injected with different combinations of ENaC subunits and with increasing amounts of γENaC cRNA. Only in the presence of the three ENaC subunit was αENaC and γENaC cleavage observed and only under these conditions was inhibition of S protein cleavage observed. With increasing amounts of γENaC more cleavage of αENAC was observed but less cleavage of $S^{WT}$ protein occurred. (D) Results of quantitation of the band corresponding to the cleaved form of $S^{WT}$ of the immunoblots represented in C. Data are mean ± SEM, *$p < 0.05$, **$p < 0.01$ vs. the only $S^{WT}$ group, n = 4. ANOVA followed by Tukey tests were performed.

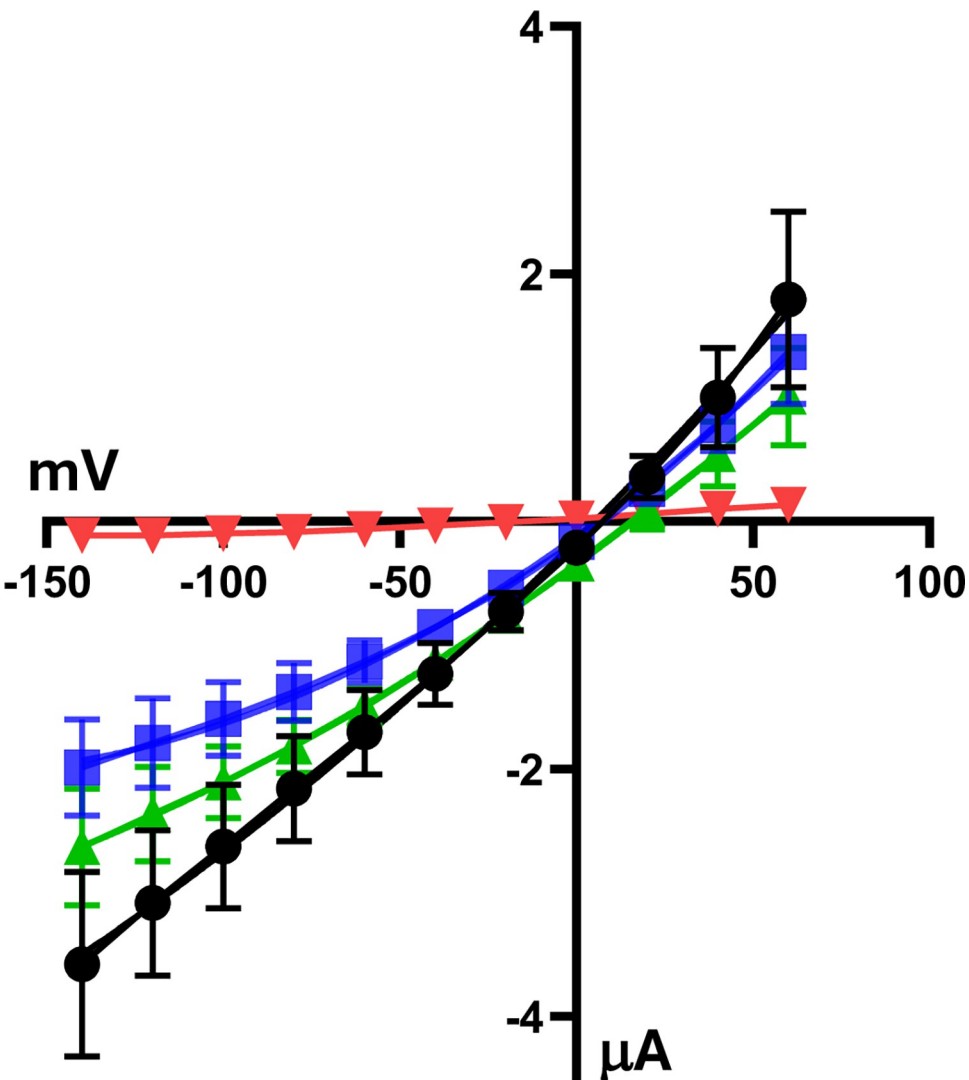

**Fig 2. ENaC activity is decreased in oocytes expressing wild type S protein.** I/V plots of amiloride-sensitive currents observed in oocytes injected with water (red), oocytes expressing ENaC heterotrimers (black), and oocytes expressing ENaC heterotrimers with wild type (blue) or mutant (green) S protein. Average currents from individual oocytes (n>6) were calculated for each experiment and averages of four experiments were plotted in the figure. Two-way ANOVA and Tukey post hoc tests were performed, showing statistical significance between the ENaC group and the ENaC + S$^{WT}$ group (Table 1 in S1 File).

also been previously tested [30]. Viral presence in lung samples was confirmed by immunofluorescence (Fig 4 in S1 File) and, as previously reported [27], infection was severe, with clearly severe lung damage. Analysis of ENaC cleavage through Western blot was not possible due to the restrictions imposed by the BSL-3 facility handling the release of tissue samples from SARS-CoV2 infected mice. Co-localization of immunofluorescent staining of γENAC with Surfactant protein A (type II pneumocytes cell marker) revealed presence of γENaC in type II pneumocytes in both, control and SARS-Cov-2 infected mice. No appreciable difference, however, in γENaC abundance between the two groups of mice could be uncovered by this method.

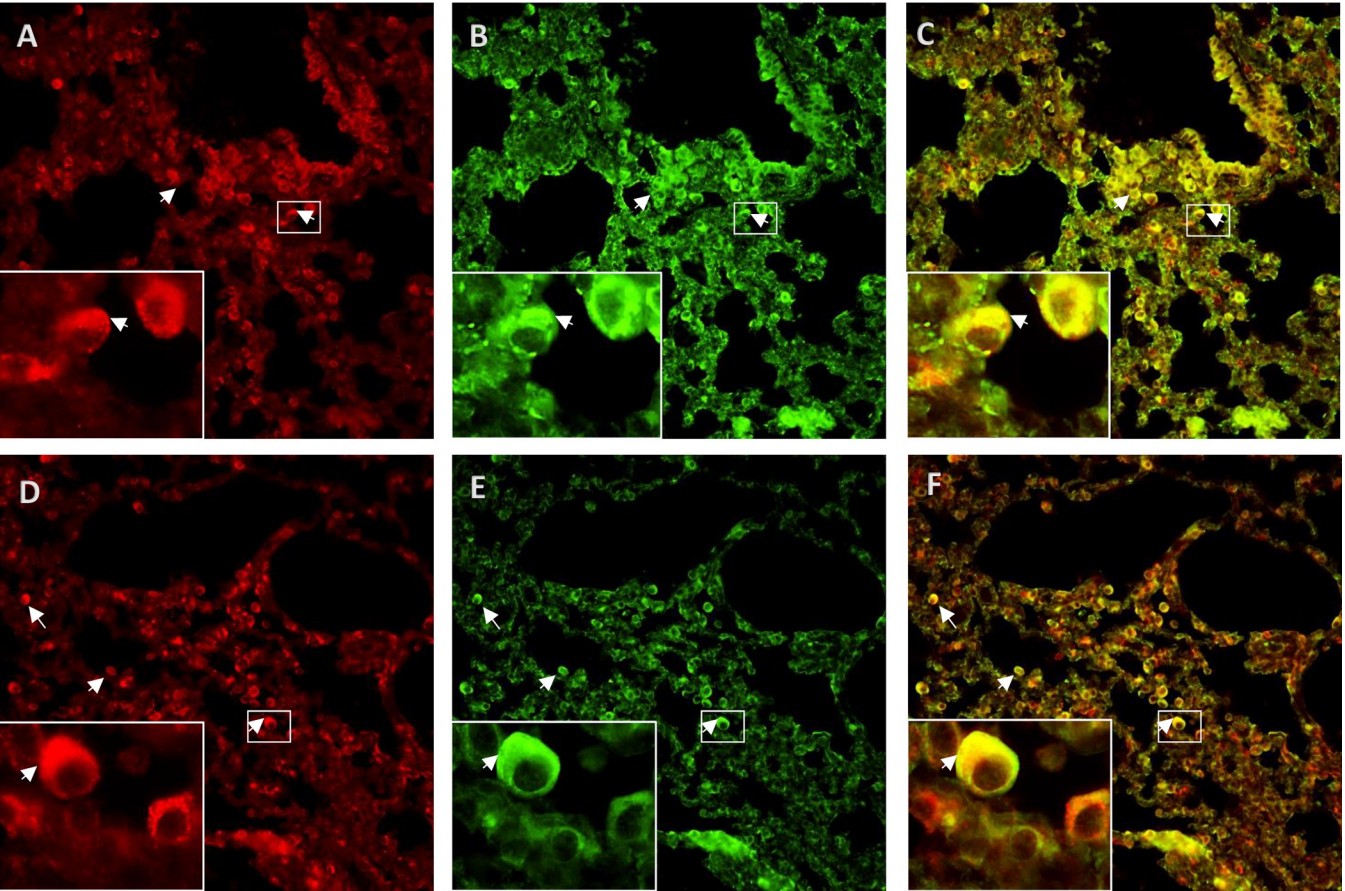

**Fig 3. Immunostaining of lungs from a control mouse and a mouse infected with SARS-CoV-2.** Immunofluorescence staining for γENaC (red), Surfactant protein A (SPA, green) and their colocalization in the merged image on the right (yellow) in lung sections of an uninfected WT C57Bl6 mouse (upper panels) and a k18hACE2 mouse infected with SARS-CoV-2 (6 dpi) (lower panels). White arrows indicate examples of γENaC staining (panels A & D) and SPA staining (panels B & E) and colocalization of γENaC and SPA in the alveoli in type II pneumocytes. All pictures were taken at 40x magnification. The insets show areas of colocalization in more detail.

## Discussion

The in silico analysis by Anand et al. showed that the furin cleavage site of SARS-CoV-2 S protein was identical to that of the αENaC furin cleavage site [9]. Since then some authors have speculated that the competition between the S protein and ENaC for furin-mediated cleavage (see Fig 4), would lead to reduced alveolar fluid clearance and alveolar-capillary barrier dysfunction. This would provide possible pathophysiological mechanisms contributing to the development of lung edema in cases of severe COVID-19 [9,14,21–24].

Here we present experimental evidence showing that such competition actually occurs in a cellular model and that this competition leads to reduced ENaC activity in vitro. Our data shows that the observed effect is bidirectional; that is, S protein prevents αENaC cleavage and ENaC prevents S protein cleavage, strengthening the idea that the effect on cleavage is indeed due to competition of both proteins for the proteolytic enzyme. In addition, the observation that the negative effect of the S protein on αENaC cleavage is slightly weakened when the cleavage site of S is mutated (in the S$^{\Delta PRRA}$ protein) further supports this hypothesis. Moreover, Grant and Lester have shown that injection in oocytes of the S protein mRNA 24 h after injection of the ENaC mRNA does not result in the inhibition of ENaC activity, suggesting

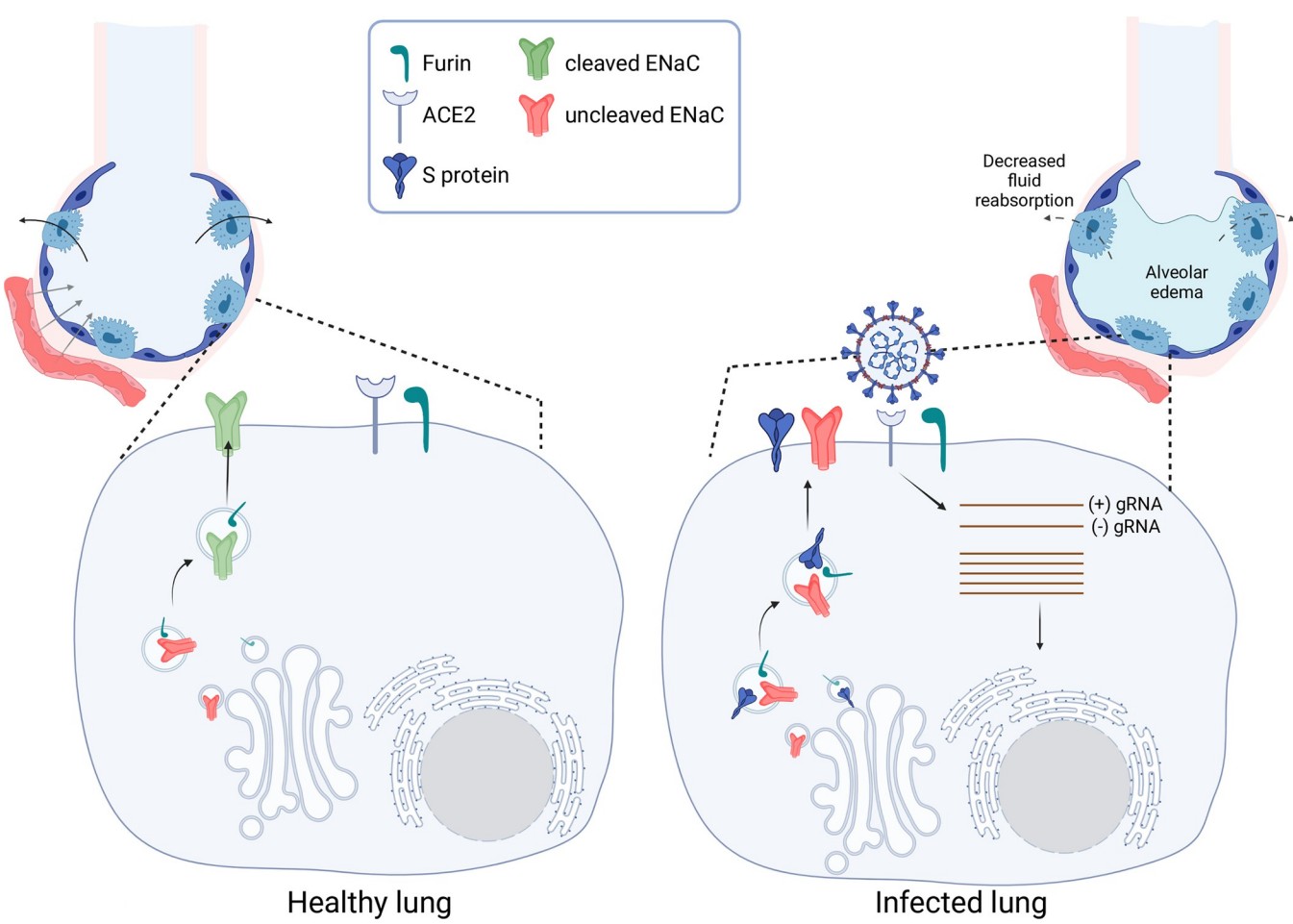

**Fig 4. Schematic representation of the proposed mechanism by which S protein expression may affect ENaC activity.** In healthy alveoli (left), ENaC in type II pneumocytes contributes to fluid reabsorption from the alveoli to allow adequate conditions for gas exchange to occur. ENaC processing by furin-like proteases during its passage through the trans-Golgi network is necessary for achieving full channel activity. When type II pneumocytes are infected with SARS-CoV-2 (right), high levels of S protein are produced in these cells. S protein and ENaC then may compete for processing enzymes. Decreased ENaC processing may cause decreased channel activity in these cells, ultimately contributing to alveolar fluid accumulation. Created with Biorender.com.

that the inhibitory effect is due to affectation of an early step in the processing and/or trafficking of the channel to the plasma membrane [29]. Interestingly, in our hands, the negative effect on αENaC cleavage was only slightly prevented with the mutation of the S protein cleavage site. One possible explanation for this is that additional residues of the S protein may be involved in protease binding that could explain the competition that is observed with the mutant. Supporting this, a docking simulation of S protein binding to furin suggests that additional residues of the S protein are involved in binding to the protease [31]. Alternatively, the S protein may inhibit ENaC cleavage by another mechanism other than competition. Our work do not address the nature of the protease responsible for the observed proteolytic events, as not only furin but also other proteases have been proposed to participate in S protein and ENaC cleavage [32]. Thus, in vivo, competition may not necessarily involve furin, but other related proteases like plasmin.

It must be noted that other mechanisms for ENaC inhibition by SARS-CoV2 have been proposed. For instance, it has been proposed that activation of Protein Kinase C (PKC), which is a known inhibitor of ENaC [33–36], may be responsible for channel inhibition by the S

protein of SARS-CoV [37]. Grant and Lester have recently shown that inhibition of PKC with the Gö-6976 inhibitor did not prevent the inhibitory effect of the S protein of SARS-CoV2 on ENaC activity in *X. laevis* oocytes [29]. However, PKC activation cannot be excluded as a relevant mechanism for ENaC inhibition in vivo since oocytes lack hACE2, which has been proposed to be an upstream mediator of this effect [14]. For instance, Romero et al. have shown that treatment with the recombinant Receptor Binding Domain of the S protein can induce ENaC inhibition in human lung microvascular endothelial cells and this has been proposed to contribute to lung vasculature disfunction [14]. This inhibition correlates with a reduction in ACE2 surface expression and generation of reactive oxygen species. Thus, it was proposed that the resulting shift in the hACE2/hACE1 balance promotes angiotensin 2 generation that activates PKC, which in turns activates NOX2, promoting the generation of reactive oxygen species. Thus, ENaC activity may be affected by several mechanisms during SARS-CoV2 infection.

As an initial approach to investigate the relevance of our hypothesis in vivo, we analyzed the expression levels of γENaC though immunofluorescent staining in samples from SARS-CoV2-infected mice. K18hACE2 transgenic mice were used, which is a frequently used mouse model in SARS-CoV-2 research that has been available since 2007 [38]. It expresses the SARS-CoV-2 receptor (full length human angiotensin-converting enzyme 2 [hACE2]) essential for viral cell entry [39] under keratin 18 promoter which directs its expression to epithelia, including airway epithelia [38]. This model develops a rapidly lethal infection after intranasal inoculation with the ancestral SARS-Cov-2 strain [27,40]. We did not observe clear differences between infected and non-infected mice. Assessment of ENaC cleavage using western blots, however, could not be done since infected tissue could not be released from the level 3 facility for safety reasons. Thus, these results should be taken with reserve given that immunofluorescent staining has several limitations. For instance, 1) immunofluorescent microscopy might not be sensitive enough to detect small differences, 2) this technique cannot discern between the cleaved and uncleaved forms of ENaC, 3) although in Western blots the antibody gives a clearly specific signal [30] (Figs 2 and 3 in S1 File), in immunofluorescent staining it is more difficult to distinguish signal from noise in the absence of a knockout control. Further research will be necessary to investigate the relative abundance and function of ENaC in infected lung tissue.

The furin site in SARS-CoV-2 S protein is one of the novel features of this virus that is not present in S proteins of closely related group 2b betacoronaviruses [4,6]. Recent works have shown that introduction of this cleavage site has contributed to the high pathogenicity and transmissibility of this virus [5,8]. For instance, Johnson et al. [8] developed a mutant SARS-CoV-2 virus carrying the deletion of the furin cleavage site (ΔPRRA) and infected hamsters and K18-hACE2 transgenic mice to test its infection ability and pathogenicity. In hamsters, despite similar viral titers, animals infected with the mutant virus presented minimal weight loss and no disease in contrast to what was observed in animals infected with the wild type virus. In mice, slightly decreased viral titers were observed in mice infected with the mutant virus at day 2 post-infection, but similar titers were observed at 7 days post infection. Despite this, weight loss and several parameters of pulmonary function were significantly more affected in mice infected with the wild type virus. Although, levels of certain cytokines were higher in mice infected with the wild type virus, this was not the case for all of them, and when compared to the levels observed in mock infected animals, cytokine levels were also elevated in the mice infected with the mutant virus. Thus, in this model, mutation of the furin cleavage site of the SARS-CoV-2 S protein reduced viral pathogenicity, without significantly affecting viral replication and with only a slightly reduced inflammatory response. In the context of our observations, it is tempting to hypothesize that this reduced pathogenicity may be due, at least

in part, to a reduced ability of the mutant S protein to interfere with ENaC cleavage and channel activity in the lung. It must be mentioned that some studies have reported that an efficient furin cleavage site in SARS-CoV-2 S protein increases replication efficiency of the virus [41] and its transmissibility in the ferret model [5].

Finally, the delta SARS-CoV-2 variant that is more transmissible and pathogenic than the original variant harbors a mutation in the furin cleavage site of the S protein (P618R). It has been shown that this mutant is more efficiently cleaved than the S protein from the original virus [41]. Thus, it is tempting to hypothesize that the increased pathogenicity of this variant may have been due, at least in part related to its increased effect on ENaC cleavage and activation. Future studies will be necessary to explore these hypotheses.

## Supporting information

**S1 File. Supplementary materials, tables, and figures.**
(PDF)

## Acknowledgments

We thank Dr. Nevan J. Krogan for the clone encoding the SARS-CoV-2 S protein. German R. Magaña Avila is a doctoral student from the "Programa de Doctorado en Ciencias Biomédicas, Universidad Nacional Autónoma de México (UNAM)" and received a fellowship from CONACYT (CVU 942671).

## Author Contributions

**Conceptualization:** Germán Ricardo Magaña-Ávila, María Castañeda-Bueno.

**Formal analysis:** Germán Ricardo Magaña-Ávila, María Castañeda-Bueno.

**Funding acquisition:** Daniel Batlle, Gerardo Gamba, María Castañeda-Bueno.

**Investigation:** Germán Ricardo Magaña-Ávila, Erika Moreno, Consuelo Plata, Héctor Carbajal-Contreras, Adrian Rafael Murillo-de-Ozores, Kevin García-Ávila, Norma Vázquez, Maria Syed, Jan Wysocki, María Castañeda-Bueno.

**Methodology:** Germán Ricardo Magaña-Ávila, Consuelo Plata, Maria Syed, Jan Wysocki, Daniel Batlle, María Castañeda-Bueno.

**Project administration:** Daniel Batlle, Gerardo Gamba, María Castañeda-Bueno.

**Resources:** Norma Vázquez.

**Supervision:** María Castañeda-Bueno.

**Validation:** Daniel Batlle, Gerardo Gamba, María Castañeda-Bueno.

**Visualization:** Germán Ricardo Magaña-Ávila, Maria Syed, Jan Wysocki, Daniel Batlle, María Castañeda-Bueno.

**Writing – original draft:** Germán Ricardo Magaña-Ávila, Daniel Batlle, María Castañeda-Bueno.

**Writing – review & editing:** Germán Ricardo Magaña-Ávila, Erika Moreno, Consuelo Plata, Maria Syed, Jan Wysocki, Daniel Batlle, Gerardo Gamba, María Castañeda-Bueno.

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
