## [Decision Letter · Decision Letter 0]

28 Dec 2023

PONE-D-23-35404Effect of SARS-CoV-2 S protein on the proteolytic cleavage of the Epithelial Na+ Channel ENaCPLOS ONE

Dear Dr. BUENO,

Thank you for submitting your manuscript to PLOS ONE. After careful consideration, we feel that it has merit but does not fully meet PLOS ONE’s publication criteria as it currently stands. Therefore, we invite you to submit a revised version of the manuscript that addresses the points raised during the review process. Please submit your revised manuscript by Feb 11 2024 11:59PM. If you will need more time than this to complete your revisions, please reply to this message or contact the journal office at plosone@plos.org. Please include the following items when submitting your revised manuscript:A rebuttal letter that responds to each point raised by the academic editor and reviewer(s). You should upload this letter as a separate file labeled 'Response to Reviewers'.A marked-up copy of your manuscript that highlights changes made to the original version. You should upload this as a separate file labeled 'Revised Manuscript with Track Changes'.An unmarked version of your revised paper without tracked changes. You should upload this as a separate file labeled 'Manuscript'.

We look forward to receiving your revised manuscript.

Kind regards,

Michael Bader

Academic Editor

PLOS ONE

Journal Requirements:

2. Please expand the acronym “CONAHCyT” (as indicated in your financial disclosure) so that it states the name of your funders in full.

"The work was supported by grants No. 101720 and A1-S-8290 from Conacyt Mexico to M-CB and GG, respectively. German R. Magaña Avila is a doctoral student from the “Programa de Doctorado en Ciencias Biomédicas, Universidad Nacional Autónoma de México (UNAM)” and received a fellowship from CONACYT (CVU 942671).

We acknowledge the support of a gift from the Joseph and Bessie Feinberg Foundation, a National Institutes of Health grant (1R21 AI166940-01) and a the Northwestern University Clinical and Translational Sciences Institute (NUCATS) COVID-19 Collaborative Innovation Award to DB."

"We thank Dr. Nevan J. Krogan for the clone encoding the SARS-CoV-2 S protein.  The work was supported by grants No. 101720 and A1-S-8290 from CONAHCyT, Mexico to M-CB and GG, respectively. German R. Magaña Avila is a doctoral student from the “Programa de Doctorado en Ciencias Biomédicas, Universidad Nacional Autónoma de México (UNAM)” and received a fellowship from CONACYT (CVU 942671).

We acknowledge the support of a gift from the Joseph and Bessie Feinberg Foundation, a National Institutes of Health grant (1R21 AI166940-01) and a the Northwestern University Clinical and Translational Sciences Institute (NUCATS) COVID-19 Collaborative Innovation Award to DB."

"The work was supported by grants No. 101720 and A1-S-8290 from Conacyt Mexico to M-CB and GG, respectively. German R. Magaña Avila is a doctoral student from the “Programa de Doctorado en Ciencias Biomédicas, Universidad Nacional Autónoma de México (UNAM)” and received a fellowship from CONACYT (CVU 942671).

We acknowledge the support of a gift from the Joseph and Bessie Feinberg Foundation, a National Institutes of Health grant (1R21 AI166940-01) and a the Northwestern University Clinical and Translational Sciences Institute (NUCATS) COVID-19 Collaborative Innovation Award to DB."

"M. Castañeda reports research funding from CONAHCyT, Mexico (grant 101720). G. Gamba reports research funding from CONAHCyT, Mexico (grant A1-S-8290). D. Batlle and J. Wysocki are coinventors of patents entitled “Active low molecular weight variants of angiotensin converting enzyme 2 (ACE2), “Soluble ACE2 variants and uses therefor.” D. Batlle is founder of Angiotensin Theraputics, Inc. D. Batlle  has received consulting fees from Advience and Traveri, all unrelated to this work. During these studies, D. Batlle received unrelated support from the National Institute of Diabetes and Digestive and Kidney Diseases (grant R01DK104785) and from a grant from AstraZeneca. D. Batlle also reports research funding from the Feinberg Foundation; J. Wysocki reports being a scientific advisor for Angiotensin Therapeutics, Inc. All other authors have nothing to disclose."

Reviewers' comments:

Reviewer's Responses to Questions

**Comments to the Author**

1. Is the manuscript technically sound, and do the data support the conclusions?

Reviewer #1: No

Reviewer #2: Yes

Reviewer #3: Partly

2. Has the statistical analysis been performed appropriately and rigorously? 

Reviewer #1: No

Reviewer #2: Yes

Reviewer #3: No

3. Have the authors made all data underlying the findings in their manuscript fully available?

Reviewer #1: No

Reviewer #2: Yes

Reviewer #3: No

4. Is the manuscript presented in an intelligible fashion and written in standard English?

Reviewer #1: Yes

Reviewer #2: Yes

Reviewer #3: Yes

5. Review Comments to the Author

Reviewer #1: In this manuscript by Magaña-Avila et al., the authors investigate whether a competition exists between SARS-CoV2 Spike protein and the alpha subunit of ENaC for cleavage by furin, using a Xenopus laevis coexpression system in vitro (WB and patch clamp) and ENaC-gamma subunit expression in immunofluorescence of lungs from SARS-CoV2-infected K18hACE2 mice. The authors conclude that a bi-directional competition exists for furin cleavage between S protein and ENaC-alpha and that this represents a main mechanism for SARS-CoV2-induced ENaC dysfunction.

Major comments.

1. Data presentation is very poor, in view of a complete lack of detail and of essential controls. As such, Western blots do not include MW markers or loading controls. The immunofluorescence images lack antibody isotype controls and quantification.

2. The conclusion of the authors is not supported by the data provided, rather the opposite is true. An S protein mutant with a deficient furin cleavage site inhibits ENaC-alpha cleavage (MW of band not described!) nearly with the same efficacy as wt S protein (Fig. 1). The described inhibition of S protein cleavage by the presence of ENaC-alpha is not clear and the lane numbers are not indicated.

3. Patch clamp data, which should include an amiloride control, only show a minor decrease in ENaC currents due to co-expression with S protein and only for certain voltages. Here mutant S protein does not affect the currents although in Fig 1 its co-expression clearly inhibits ENaC-alpha cleavage. As such, there is a complete disconnect between data in Figs 1 and 2.

4. The co-expression system is artificial since it ignores the signal transduction initiated by binding of S protein to ACE2 in the lungs. ACE2 downregulation upon S protein binding impairs the ACE/ACE2 balance and can activate PKC, a known inhibitor of ENaC open probability (Bao et al., Am J Physiol Renal Physiol. 2007; Chen et al., Am J Physiol Lung Cell Mol Physiol. 2004).

Reviewer #2: In their study, Magana-Avila et al. report the competitive inhibition of cleavage of the alpha subunit of the epithelial sodiun channel ENaC by coexpression of SARS-CoV 2 spike protein in the Xenopus oocyte expression system. Moreover, the authors find that this coincides with reduced ENaC activity in TEVC experiments. Unfortunately, the authors could not present evidence for the impact in vivo since lungs from infected mice were not available for Western blot.

The study is straightforward and the results are presented in a clear and understandable fashion. I particularly like Fig 4 generating a hypothesis fort he pathophysiological impact of the findings.

I have the following suggestions:

1. Fig 1: why is there inhibition of aENaC cleavage at 0.05 spike protein expression whereas cleavage of spike protein is only detectable at 0.4 spike protein expression?

2. Fig 1: It would be interesting to see data on cleavage of gENaC as well since furin is supposed to cleave gENaC once near the N terminus. I suggest the authors to present data on the cleavage of gENaC using the same antibody as used for IF (Stressmarq SPC-405)?

3. Fig. 1: please indicate molecular size of cleaved aENaC.

4. Fig 2: please give sample traces of amiloride-sensitive currents for all groups at e.g. -140 mV holding potential

5. Fig 3: the background seems to high. Can the authors optimize the conditions by diluting the antibody or improving antigen retrieval? The authors should also give a view with a higher magnification as inset.

Reviewer #3: I have reviewed the manuscript Effect of SARS-CoV-2 S protein on the proteolytic cleavage of the Epithelial Na+ Channel ENaC by Bueno et al.

Authors hypothesize that the S-protein competes for furin clevage with the alpha (and gamma) ENaC subunit in lung and that this could contribute to inactivate ENaC and promote fluid accumulation. The hypothesis is tested in vitro in the Xenopus expression system and in vivo in mice exposed to SARS Cov-2 virus. Oocytes were injected with mRNA for all EnaC subunits and with Sprotein.

Alpha ENaC cleavage and inward amiloride-sensitive currents were attenuated by S-protein overexpression but not in S-protein with mutated furin-cleavage site. Only immunofluorescence was possible in infected mouse lungs. Data indicate that overexpression of S-protein engages furin and leaves alpha ENaC uncleaved.

The idea is good and novel and well presented in the introduction. I have some suggestions that authors could consider in order to improve their manuscript.

1. From which species were the mRNAs that were injected?

2. Why was furin not overexpressed and where does the proposed reaction occur? In the biosynthesis pathway or on the surface? I realize there is a schematic drawing but no data are presented on this issue.

3. Is the amount of furin always the same or in other words, does the alleged competition not also depend on the amount of furin?

4. Can it be excluded that other inherent/endogenous proteases contribe to cleavage?

5. It would strengthen the data to knock out furin in oocytes or at least demonstrate furin.

6. Why was an accepted protease inhibitior not used as a positive control, i.e. aprotinin or alike to test that a similar inhibition as with S-protein overexpression was observed and to map the part of the reaction taking place on the cell surface.

7. How do you compare densitometry across gels on the immunoblots – this should be described better(/in detail.

8. In figure 1, please state the expected migratory pattern of each protein species and how it compares with actual migration and place independent molecular weight markers. A C-terminal antibody against human αENaC would show proteins migrating putatively at 74, 51, and 48 kDa (non-glycosylated). Was that the case ?

9. Authors express alpha, beta and gamma ENaC -but I can only find immunoblotting data on alpha. Why are especially gamma ENaC not evaluated since it also depends on furin for cleavage and an extra cleavage to gain full activity in the intact channel.

10. Frankly, n=3 does not allow meaningful statistical evaluation (fig 1). Statistical methods are not described except for fig 2 data (or I cannot find it). Please state in figure legends.

11. What is “K18hACE2 transgenic mice” -please provide details. How many mice were used in total and how many died before inclusion- these parameters should be reported

12. It is a pity that fresh lung tissue from the mice was not available for immunoblotting, however it would have some value to validate immunofluorescence by running immunoblots on non-infected control mice to demonstrate that the antigen is significantly present in the adult mouse lung. And compared to kidney.

13. In contrast to the immunoblotting where only alpha ENaC data are shown, in immunofluorescence only gammaENaC is shown. Why? Please add data on alpha. In the legend to fig 3 it is stated “ENaC” but not alpha or gamma. Please clarify.

14. The number of the protocol approving the experiments and the date should be given.

15. The magnificantion of the immunofluorescence and/or the resolution in my cope does not allow to see much detail. Could higher magnificantion be shown.

16. Please delete the speculative parts of the discussion on dexamethasone, it is tangential, not tested in the study and although interesting, not relevant in the present context.

6. PLOS authors have the option to publish the peer review history of their article (what does this mean?). If published, this will include your full peer review and any attached files.

Reviewer #1: **Yes: **Rudolf Lucas

Reviewer #2: No

Reviewer #3: No

---

## [Author Response · Author response to Decision Letter 0]

11 Feb 2024

RESPONSE TO REVIEWERS

Reviewer #1: In this manuscript by Magaña-Avila et al., the authors investigate whether a competition exists between SARS-CoV2 Spike protein and the alpha subunit of ENaC for cleavage by furin, using a Xenopus laevis coexpression system in vitro (WB and patch clamp) and ENaC-gamma subunit expression in immunofluorescence of lungs from SARS-CoV2-infected K18hACE2 mice. The authors conclude that a bi-directional competition exists for furin cleavage between S protein and ENaC-alpha and that this represents a main mechanism for SARS-CoV2-induced ENaC dysfunction.

Major comments.

1. Data presentation is very poor, in view of a complete lack of detail and of essential controls. As such, Western blots do not include MW markers or loading controls. The immunofluorescence images lack antibody isotype controls and quantification.

R: We are sorry that the reviewer got this impression. We have now modified the manuscript in an attempt to improve the reviewer´s perception of the manuscript. We have included loading controls and additional MW markers to those originally included. 

We have also included as supplementary information some validation data for the gamma ENaC antibody used for immunofluorescence (Supplementary figures 2 and 3). In Western blot with kidney samples we were able to show that the antibody recognizes only two bands of the expected molecular weights for the uncleaved and cleaved forms of gamma ENaC (Figure S2). Importantly, the expected increase in the cleaved form of gamma ENaC was observed in samples from animals treated with amiloride, which develop severe volume depletion and hyperkalemia, and thus, in which ENaC activation and gamma ENaC cleavage is expected. This shows that the antibody is able to specifically recognize gamma ENaC in WBs with kidney tissue samples. 

In addition, in immunofluorescent staining of kidney samples, strong tubular staining with very little background fluorescence was observed (Figure S2). Signal was localized to principal cells of the nephron that were negative for ATPase B1 staining, which marks intercalated cells in connecting tubules and collecting ducts. 

In lung tissue (Figure 3 and Figure S3), signals of higher and lower intensity were observed. This pattern was similar to that observed with an antibody against the Surfactant protein A (a marker of type 2 pneumocytes, where ENaC is known to be expressed) (Figure 3). Similar staining was observed in lungs from non-infected WT mice and K18 infected mice. Negative control was done by omission of primary gamma ENaC antibody in the IF procedure that was substituted by incubation with the diluent only (Figure S3).

Finally, it is worth mentioning that the used gamma ENaC antibody has been previously validated by others (1).

2. The conclusion of the authors is not supported by the data provided, rather the opposite is true. An S protein mutant with a deficient furin cleavage site inhibits ENaC-alpha cleavage (MW of band not described!) nearly with the same efficacy as wt S protein (Fig. 1). The described inhibition of S protein cleavage by the presence of ENaC-alpha is not clear and the lane numbers are not indicated.

R: It is true that, as the reviewer states, the S protein mutant with a deficient furin cleavage site still appears to significantly prevent alpha ENaC cleavage. It is only at the lowest S protein concentration tested (0.05 ug/ul) that a difference between the wild type S protein and the mutant S protein is appreciated. In the discussion, we speculate that perhaps the mutant S protein may still be able to bind with slightly less affinity to the protease and in this way exert a competitive effect that prevents ENaC cleavage. We cannot rule out however, that other explanations may exist. For instance, the S protein may inhibit ENaC cleavage by another mechanism other than competition. However, we think that the observation that ENaC´s presence also prevents S protein cleavage supports the idea of competition. Interestingly, supporting the idea that the S protein affects an early step in the channel’s processing and /or trafficking to the membrane, it has been shown in oocytes that injecting the mRNA encoding for the S protein 24 h after injection of ENaC mRNA does not result in inhibition of ENaC activity (2). This evidence has now been cited in the manuscript and the above discussion has now been included. 

We think that our data does indeed show clear inhibition of S protein cleavage by ENaC co-expression. We emphasize this in the legend of figure 1: “It is noteworthy that proteolytic processing of SWT was clearly prevented in the presence of ENaC coexpression (bands corresponding to the cleaved S2 subunit observed in oocytes injected with the same amount of SWT in the absence or presence of ENaC are highlighted with boxes).” We also emphasize this in the results section: “The proteolytic processing of SWT was also appreciated in the immunoblots. As expected, this processing was not observed for the S�PRRA mutant. It is noteworthy that in the presence of ENaC, SWT processing was reduced (compare lane 2 with lane 7 of the blot in Fig.1 in which equivalent levels of SWT cRNA were injected).” We have now included the lane numbers in Figure 1 to facilitate identification of the lanes mentioned in the text. 

In addition, we now include new blots (figure 1C-D) from experiments in which the amount of S protein was kept constant and different combinations of ENaC subunits in different amounts were co-expressed. In these experiments, we confirmed that, as previously shown by others (3), it is necessary for the three ENaC subunits to be present to observe alpha ENaC and gamma ENaC cleavage. Moreover, we observed that only in the presence of the three subunits, S protein cleavage decreases and that as gamma ENaC concentration increases and more cleavage of alpha ENAC and gamma ENaC is observed, less cleavage of the S protein is observed. 

3. Patch clamp data, which should include an amiloride control, only show a minor decrease in ENaC currents due to co-expression with S protein and only for certain voltages. Here mutant S protein does not affect the currents although in Fig 1 its co-expression clearly inhibits ENaC-alpha cleavage. As such, there is a complete disconnect between data in Figs 1 and 2.

R: We apologize for the confusion. We would like to clarify that the two-electrode voltage clamp results reported already consider the amiloride control. This is, for every given voltage tested, the current values plotted are the values obtained in the absence of amiloride minus the values obtained in its presence. The current-voltage relationship of amiloride-sensitive currents that we observed is similar to that reported by others (4). Please refer to the “Two-electron voltage clamp” section in the Materials and Methods section where a description of these calculations is included. 

This comment by the reviewer actually motivated us to reanalyze our electrophysiological data. In our new analysis we now consider the average current from all oocytes in each experiment instead of considering current values from individual oocytes. We decided to do this to balance the weight given to each experiment, given that different amounts of oocytes were studied in each experiment. To evaluate if there were differences between the curves, we performed a two-way ANOVA of the repeated samples and a post-hoc Tukey test showing significative statistical difference between the ENaC and ENaC + SWT groups. 

As the reviewer points out, only a small but significant decrease in ENaC-mediated currents was observed in the presence of S protein. This is despite the apparently complete prevention of alpha ENaC cleavage shown in the blots represented in figure 1. We do not believe, however, that this is unexpected, as others have reported only partial decrease in ENaC currents with mutant channels in which the cleavage site is mutated and thus cleavage is completely prevented (3). 

Finally, it is true that we also expected to see a decrease in ENaC currents in the presence of the mutant S protein, given that this mutant also significantly prevents ENaC cleavage. We do see a tendency for the ENaC currents to be smaller in the presence of the S mutant. However, these are not statistically significant due to the data variation. Interestingly, however, others have also reported that ENaC-mediated currents decrease in the presence of the wild type S protein and that the inhibition observed is only partially prevented with a mutant S protein (2). This work is now cited in our manuscript (page 10).

 4. The co-expression system is artificial since it ignores the signal transduction initiated by binding of S protein to ACE2 in the lungs. ACE2 downregulation upon S protein binding impairs the ACE/ACE2 balance and can activate PKC, a known inhibitor of ENaC open probability (Bao et al., Am J Physiol Renal Physiol. 2007; Chen et al., Am J Physiol Lung Cell Mol Physiol. 2004).

R: We appreciate this comment that has given us the opportunity to discuss the possible role of PKC and other proposed mechanisms that may contribute to ENaC disfunction in our manuscript (see discussion section). Fortunately, the role of PKC has been previously addressed by others and Grant et. al. have shown that inhibition of PKC with Go¨-6976 do not prevent the decrease in ENaC-mediated Na+ currents in oocytes co-injected with SARS-CoV-2 S protein (2). It has also been shown that treatment with the recombinant Receptor Binding Domain of the S protein can induce ENaC inhibition in human lung microvascular endothelial cells and this has been proposed to contribute to lung vasculature disfunction (5). This inhibition correlates with a reduction in ACE2 surface expression and generation of reactive oxygen species. And given that only extracellular exposure to the Receptor Binding Domain of the S protein is necessary to induce ENaC inhibition, the mechanism of inhibition is probably unrelated to an effect on ENaC cleavage. These works are now cited in the manuscript.

We agree that in our system we may have missed other possible mechanisms for ENaC inhibition by SARS-CoV2. Nevertheless, we do not attempt to rule out other possibly involved mechanisms, rather to contribute by addressing this particular mechanism that involves defective ENaC cleavage and for which the oocyte system appears to be suitable. 

Reviewer #2: In their study, Magana-Avila et al. report the competitive inhibition of cleavage of the alpha subunit of the epithelial sodiun channel ENaC by coexpression of SARS-CoV 2 spike protein in the Xenopus oocyte expression system. Moreover, the authors find that this coincides with reduced ENaC activity in TEVC experiments. Unfortunately, the authors could not present evidence for the impact in vivo since lungs from infected mice were not available for Western blot.

The study is straightforward and the results are presented in a clear and understandable fashion. I particularly like Fig 4 generating a hypothesis for the pathophysiological impact of the findings.

I have the following suggestions:

1. Fig 1: why is there inhibition of aENaC cleavage at 0.05 spike protein expression whereas cleavage of spike protein is only detectable at 0.4 spike protein expression?

R: This is probably a misunderstanding. Cleavage of spike is detected at lower levels of spike expression. For instance, in lane number 2 of the blot, it can be seen that when spike is injected at 0.2 ug/ul cleavage is observed. However, with this same amount of spike, but in the presence of ENaC (lane #7) no cleaved spike is observed probably due to competition with ENaC. 

2. Fig 1: It would be interesting to see data on cleavage of gENaC as well since furin is supposed to cleave gENaC once near the N terminus. I suggest the authors to present data on the cleavage of gENaC using the same antibody as used for IF (Stressmarq SPC-405)?

R: We have performed blots of gamma ENaC as suggested by the reviewer. Interestingly, in these blots we observed that alpha ENaC appears to be more efficiently cleaved than gamma ENaC because the proportion of cleaved/full length gamma ENaC is higher than that observed for aENaC and spike (see for example figure 1C, lane 5, in which equal amounts of alpha ENAC and gamma ENaC were injected). We do not see, however, a clear negative effect of spike expression on gamma ENaC cleavage as that observed for alpha ENaC. This effect was only observed in some blots and at high levels of spike expression. Below you can see representative blots for both situations (cleavage prevented and not prevented). We believe this could be due to the more higher efficiency of cleavage of gamma ENaC. This is, if gamma ENaC is preferentially cleaved, then perhaps it is more difficult to observe prevention of cleavage by spike´s presence. Alternatively, cleavage of gamma ENaC by another protease could also explain these observations. These observations, however, do not rule out that gamma ENaC cleavage could be affected in vivo in infected mice under conditions in which S protein expression may be much higher than ENaC expression. 

Our observations suggest that it might be impossible to dissect if it is the presence of alpha ENaC, gamma ENaC, or both what prevents spike cleavage. In the new figure 1C-D we show that only in the presence of the three subunits, alfa and gamma ENaC are cleaved (this has also been shown by others (3)). Of note, it is also only in the presence of the 3 subunits of ENaC that the cleavage of spike is inhibited. In other words, only when cleavage of ENaC is possible, the inhibition of the cleavage of spike is observed, and as gamma ENaC concentration increases and more cleavage of alpha ENAC and gamma ENaC is observed, less cleavage of the S protein is observed.

3. Fig. 1: please indicate molecular size of cleaved aENaC.

This information has been included in the figure legend.

4. Fig 2: please give sample traces of amiloride-sensitive currents for all groups at e.g. -140 mV holding potential

R: Traces have been included in supplementary figure 1.

5. Fig 3: the background seems to high. Can the authors optimize the conditions by diluting the antibody or improving antigen retrieval? The authors should also give a view with a higher magnification as inset. 

R: Thank you for this comment. As described in response to comment 1 of the Reviewer 1, we did a careful validation of the gamma ENaC antibody that was used for immunofluorescence (IF) (Figures S3 and S4). This validation suggests that the observed signal in IFs of lung tissue is specific. We tested different antigen retrieval protocols and also IF with no antigen retrieval procedure. For the images presented in figures 3, S2 and S3, we used the protocol in which best results were obtained with kidney and lung samples. As mentioned before, to rule out non-specific staining in the lungs, negative control of lung tissue with omission of primary antibody was performed which resulted in virtually no fluorescence. This suggested that the IF signals of lower intensity observed in the lung tissue are likely not background related. 

As the reviewer suggested, we now provide insets of higher resolution in the new Figure 3.

Reviewer #3: I have reviewed the manuscript Effect of SARS-CoV-2 S protein on the proteolytic cleavage of the Epithelial Na+ Channel ENaC by Bueno et al.

Authors hypothesize that the S-protein competes for furin clevage with the alpha (and gamma) ENaC subunit in lung and that this could contribute to inactivate ENaC and promote fluid accumulation. The hypothesis is tested in vitro in the Xenopus expression system and in vivo in mice exposed to SARS Cov-2 virus. Oocytes were injected with mRNA for all EnaC subunits and with Sprotein.

Alpha ENaC cleavage and inward amiloride-sensitive currents were attenuated by S-protein overexpression but not in S-protein with mutated furin-cleavage site. Only immunofluorescence was possible in infected mouse lungs. Data indicate that overexpression of S-protein engages furin and leaves alpha ENaC uncleaved.

The idea is good and novel and well presented in the introduction. I have some suggestions that autho

---

## [Decision Letter · Decision Letter 1]

11 Mar 2024

PONE-D-23-35404R1Effect of SARS-CoV-2 S protein on the proteolytic cleavage of the Epithelial Na+ Channel ENaCPLOS ONE

Dear Dr. BUENO,

Thank you for submitting your manuscript to PLOS ONE. After careful consideration, we feel that it has merit but does not fully meet PLOS ONE’s publication criteria as it currently stands. Therefore, we invite you to submit a revised version of the manuscript that addresses the points still raised by the reviewer.

We look forward to receiving your revised manuscript.

Kind regards,

Michael Bader

Academic Editor

PLOS ONE

Reviewers' comments:

Reviewer's Responses to Questions

**Comments to the Author**

1. If the authors have adequately addressed your comments raised in a previous round of review and you feel that this manuscript is now acceptable for publication, you may indicate that here to bypass the “Comments to the Author” section, enter your conflict of interest statement in the “Confidential to Editor” section, and submit your "Accept" recommendation.

Reviewer #1: (No Response)

2. Is the manuscript technically sound, and do the data support the conclusions?

Reviewer #1: No

3. Has the statistical analysis been performed appropriately and rigorously? 

Reviewer #1: No

4. Have the authors made all data underlying the findings in their manuscript fully available?

Reviewer #1: No

5. Is the manuscript presented in an intelligible fashion and written in standard English?

Reviewer #1: Yes

6. Review Comments to the Author

Reviewer #1: In this revised version, it is appreciated that the authors tried to address the comments made by the three reviewers. However, a lot of issues remain for this reviewer with poorly controlled experiments, doubtful interpretation of results and weak consideration of alternative explanations. Here a summary:

Introduction

1. Not only type 2 but also type 1 AT express ENaC

2. COVID-ARDS is not more inflammatory than non COVID ARDS. As such, listing inflammation as the main cause of edema formation is questionable. Alveolar-capillary barrier dysfunction is also very important and is not discussed.

Results

1. As brought up before, the mutant S protein has only a two-fold reduced capacity to inhibit ENaC-alpha cleavage as compared to the WT S protein, strongly questioning the main hypothesis.

2. As brought up before, in Fig 2, it is not acceptable to just subtract the amiloride data from the rest. All voltage current tracers should be shown.

3. The fact that the mutant S protein does not significantly differ from the WT S protein in the patch clamp study, together with a lack of effect of SARS-CoV2 infection on ENaC-gamma expression in lungs in Fig. 3 highly questions the relevance of the proposed hypothesis.

4. Lack of an essential isotype control Ab in Fig. 3. Referring to one manuscript where the anti-ENaC-gamma antibody was used does not suffice.

5. The manuscript now mentions furin-like proteases throughout, but uses an S protein mutant that has a specific deletion in the furin cleavage site. This is confusing. As suggested by another reviewer, why didn't the authors overexpress furin or used furin inhibitors in their studies? This weakens the aim of the study.

6. Whereas SP-C is a specific marker for type 2 alveolar epithelial cells, SP-A, used in Fig 3, is not, since it can also be secreted by non-ciliated bronchiolar cells, submucosal gland and epithelial cells of other respiratory tissues, such as the trachea and bronchi (Carreto-Binaghi LE, Aliouat el M, Taylor ML. Surfactant proteins, SP-A and SP-D, in respiratory fungal infections: their role in the inflammatory response. Respir Res. 2016 Jun 1;17(1):66. doi: 10.1186/s12931-016-0385-9. PMID: 27250970; PMCID: PMC4888672).

Discussion

1. PKC activation is being discarded by the authors as an explanation for the ENaC- inhibitory effect of the S protein. However, a recent paper performed in primary human cells showed that the receptor binding domain of S1 -so the form after cleavage by furin- is highly efficient in inhibiting ENaC activity, albeit in a different cell type (human lung MVEC). The oocyte expression system used by the authors or by the Clark paper they refer to does not allow to investigate this, since there is no human ACE2 expression. If the cleaved S protein binds to hACE2, this increases ACE and subsequent PKC activity. As such, if the machinery to mediate S protein-induced PKC activation is missing no valid conclusions can be made.

In conclusion, the authors did not convincingly demonstrate that SARS-CoV2 S protein inhibits ENaC-alpha and gamma cleavage mainly by hijacking furin. There data rather indicate that this pathway is not important.

7. PLOS authors have the option to publish the peer review history of their article (what does this mean?). If published, this will include your full peer review and any attached files.

Reviewer #1: No

---

## [Author Response · Author response to Decision Letter 1]

2 Apr 2024

Reviewer #1: In this revised version, it is appreciated that the authors tried to address the comments made by the three reviewers. However, a lot of issues remain for this reviewer with poorly controlled experiments, doubtful interpretation of results and weak consideration of alternative explanations. Here a summary:

Introduction

1. Not only type 2 but also type 1 AT express ENaC

We apologize for this confusion. What we meant to say in the introduction, is that it is in type 2 pneumocytes where the channel is co-expressed with ACE2 and furin. The text has now been modified to avoid this confusion. 

2. COVID-ARDS is not more inflammatory than non-COVID ARDS. As such, listing inflammation as the main cause of edema formation is questionable. Alveolar-capillary barrier dysfunction is also very important and is not discussed.

We thank the reviewer for this comment that has given us the opportunity to mention this additional mechanism by which ENaC inhibition may contribute to the pathophysiology of COVID-19 in the introduction and discussion sections. 

Results

1. As brought up before, the mutant S protein has only a two-fold reduced capacity to inhibit ENaC-alpha cleavage as compared to the WT S protein, strongly questioning the main hypothesis.

We agree with the reviewer as we also expected a lack of inhibition of cleavage with the mutant S protein. Unfortunately, as with many biological phenomena, this does not seem to be an all or nothing effect on ENaC cleavage. However, we do not believe that this questions the idea that S protein prevents ENaC cleavage, which is the main finding of our manuscript. 

In the manuscript, we mention that inhibition of cleavage by the mutant S protein may be due to preserved binding, albeit with slightly less affinity, of this mutant protein to the protease and in this way exert a competitive effect that prevents ENaC cleavage. Supporting this, a docking simulation of S protein binding to furin has shown that additional residues of the S protein could be involved in binding to the protease (1). We cannot rule out however, that other explanations may exist. For instance, the S protein may inhibit ENaC cleavage by another mechanism other than competition. However, the observation that ENaC´s presence also prevents S protein cleavage supports the idea of competition. Interestingly, as mentioned before, suggesting that S protein affects an early step in the channel’s processing and /or trafficking to the membrane, it has been shown in oocytes that injecting the mRNA encoding for the S protein 24 h after injection of ENaC mRNA does not result in inhibition of ENaC activity (2). 

2. As brought up before, in Fig 2, it is not acceptable to just subtract the amiloride data from the rest. All voltage current tracers should be shown.

We have now included these data in supplementary figure 1. 

3. The fact that the mutant S protein does not significantly differ from the WT S protein in the patch clamp study, together with a lack of effect of SARS-CoV2 infection on ENaC-gamma expression in lungs in Fig. 3 highly questions the relevance of the proposed hypothesis.

As mentioned in point 1 (of comments to results section), the observation that the mutations in the cleavage site of S protein do not fully prevent the negative effect on cleavage and activity of ENaC may have multiple explanations, some of which are discussed above. It is correct that the results presented in figure 3 question the relevance of the hypothesis in vivo. However, this is clearly stated in the manuscript and we believe that publishing this negative result will be valuable to the scientific community. 

4. Lack of an essential isotype control Ab in Fig. 3. Referring to one manuscript where the anti-ENaC-gamma antibody was used does not suffice.

The manuscript to which we refer, is not only a manuscript in which the antibody was used, but is a manuscript in which the main purpose was to validate the ENaC antibodies. In addition, we provide in our manuscript substantial validation data showing that, in our hands, the antibody is able to specifically recognize gamma ENaC. We think that the ideal control to validate the antibody for immunofluorescence would be a knockout animal, however, given that we don´t have access to this model, this is beyond our capabilities. To address this limitation we have included a comment in the discussion section. Nevertheless, if the reviewer is not comfortable with this part of the manuscript we could move these data to supplemental information section given that it is not an essential part of the manuscript as it does not directly address whether ENaC cleavage is affected by SARS-CoV2 infection. 

5. The manuscript now mentions furin-like proteases throughout, but uses an S protein mutant that has a specific deletion in the furin cleavage site. This is confusing. As suggested by another reviewer, why didn't the authors overexpress furin or used furin inhibitors in their studies? This weakens the aim of the study.

It must be clarified that other furin-like proteases are expected to cleave ENaC and S protein in the furin site. We decided to mention furin-like proteases instead of just furin to leave open the possibility that the cleavage might be mediated by another furin-like protease. As such, we deemed it futile to specifically demonstrate that furin is the responsible protease for ENaC and S protein cleavage in our system. Whichever the protease, it remains true that cleavage of these proteins is inhibited by one another.

6. Whereas SP-C is a specific marker for type 2 alveolar epithelial cells, SP-A, used in Fig 3, is not, since it can also be secreted by non-ciliated bronchiolar cells, submucosal gland and epithelial cells of other respiratory tissues, such as the trachea and bronchi (Carreto-Binaghi LE, Aliouat el M, Taylor ML. Surfactant proteins, SP-A and SP-D, in respiratory fungal infections: their role in the inflammatory response. Respir Res. 2016 Jun 1;17(1):66. doi: 10.1186/s12931-016-0385-9. PMID: 27250970; PMCID: PMC4888672).

We thank the reviewer for bringing this up. However, we did co-staining studies in the context of alveoli and not the airway tissues, such as bronchioles, bronchi, or trachea where SP-A can also be present, as the reviewer appropriately pointed out. Since those structures are easily discernable from the alveolar space under microscopy, we think that this should not affect the interpretation of our co-staining images that were focused on the alveolar space and not on airways. We pointed out this in the Methods section.

Discussion

1. PKC activation is being discarded by the authors as an explanation for the ENaC- inhibitory effect of the S protein. However, a recent paper performed in primary human cells showed that the receptor binding domain of S1 -so the form after cleavage by furin- is highly efficient in inhibiting ENaC activity, albeit in a different cell type (human lung MVEC). The oocyte expression system used by the authors or by the Clark paper they refer to does not allow to investigate this, since there is no human ACE2 expression. If the cleaved S protein binds to hACE2, this increases ACE and subsequent PKC activity. As such, if the machinery to mediate S protein-induced PKC activation is missing no valid conclusions can be made.

In the discussion, we now add a sentence mentioning PKC activation as one of the possible mechanisms for ENaC inhibition and cite the paper where the idea came from (Front. Immunol. 14:1241448) (3). In addition, it is important to mention that it is now explicitly stated in our manuscript, that our model cannot explore this possible alternative mechanism due to absence of ACE2 expression. Nevertheless, this model seems to be ideal to explore the competition of S protein and ENaC for cleavage and this only would represent an additional mechanism that could explain ENaC inhibition in SARS-COV2 infection. 

In conclusion, the authors did not convincingly demonstrate that SARS-CoV2 S protein inhibits ENaC-alpha and gamma cleavage mainly by hijacking furin. There data rather indicate that this pathway is not important.

We are sorry that the reviewer got this impression. In our manuscript we clearly show that alpha ENaC cleavage is prevented by S protein co-expression and that the opposite is also true. We also show that ENaC activity is inhibited by S protein co-expression as others have already shown. We also mention other pathways that are involved in ENaC inhibition by S protein and that are not mutually exclusive with our proposed mechanism. Finally, although our in vivo data does not support the mechanism that we propose, this is clearly stated in the manuscript and we believe that this negative result will be valuable to the scientific community and will motivate further studies. There has been a lot of speculation in the scientific literature around this proposed mechanism and we believe that this manuscript will significantly contribute to this discussion.

---

## [Editor Report · Decision Letter 2]

4 Apr 2024

Effect of SARS-CoV-2 S protein on the proteolytic cleavage of the Epithelial Na+ Channel ENaC

PONE-D-23-35404R2

Dear Dr. BUENO,

We’re pleased to inform you that your manuscript has been judged scientifically suitable for publication and will be formally accepted for publication once it meets all outstanding technical requirements.

Kind regards,

Michael Bader

Academic Editor

PLOS ONE